# Planimetric and Volumetric Brainstem MRI Markers in Progressive Supranuclear Palsy, Multiple System Atrophy, and Corticobasal Syndrome. A Systematic Review and Meta-Analysis

**Maria-Evgenia Brinia, Ioanna Kapsali, Nikolaos Giagkou and Vasilios C. Constantinides ***

First Department of Neurology, School of Medicine, National and Kapodistrian University of Athens, Eginition Hospital, 11528 Athens, Greece; mariaevgeniabr@gmail.com (M.-E.B.); ioannakapsali@ymail.com (I.K.)
* Correspondence: vconstan@med.uoa.gr

**Abstract:** Background: Various MRI markers—including midbrain and pons areas ($M_{area}$, $P_{area}$) and volumes ($M_{vol}$, $P_{vol}$), ratios ($M/P_{area}$, $M/P_{vol}$), and composite markers (magnetic resonance imaging Parkinsonism Indices 1,2; MRPI 1,2)—have been proposed as imaging markers of Richardson's syndrome (RS) and multiple system atrophy–Parkinsonism (MSA-P). A systematic review/meta-analysis of relevant studies aiming to compare the diagnostic accuracy of these imaging markers is lacking. Methods: Pubmed and Scopus were searched for studies with >10 patients (RS, MSA-P or CBS) and >10 controls with data on $M_{area}$, $P_{area}$, $M_{vol}$, $P_{vol}$, $M/P_{area}$, $M/P_{vol}$, MRPI 1, and MRPI 2. Cohen's *d*, as a measure of effect size, was calculated for all markers in RS, MSA-P, and CBS. Results: Twenty-five studies on RS, five studies on MSA-P, and four studies on CBS were included. Midbrain area provided the greatest effect size for differentiating RS from controls (Cohen's *d* = −3.10; *p* < 0.001), followed by $M/P_{area}$ and MRPI 1. MSA-P had decreased midbrain and pontine areas. Included studies exhibited high heterogeneity, whereas publication bias was low. Conclusions: Midbrain area is the optimal MRI marker for RS, and pons area is optimal for MSA-P. $M/P_{area}$ and MRPIs produce smaller effect sizes for differentiating RS from controls.

**Keywords:** Richardson's syndrome; progressive supranuclear palsy; multiple system atrophy; corticobasal syndrome; planimetry; volumetry; midbrain; pons; magnetic resonance Parkinsonism index; systematic review; meta-analysis

## 1. Introduction

Atypical Parkinsonian disorders (APD) is a term used to describe three rare neurodegenerative Parkinsonian disorders, namely progressive supranuclear palsy (PSP), multiple system atrophy (MSA), and corticobasal syndrome (CBS) [1–3]. PSP exhibits significant phenomenological heterogeneity, with Richardson's syndrome (RS) being the most common syndrome, characterized by supranuclear gaze palsy and early postural instability. MSA presents with two distinct syndromes, with predominant parkinsonian (MSA-P) or cerebellar (MSA-C) symptomatology. Despite the presence of distinct clinical features in RS, MSA-P, and CBS, misdiagnosis is common, particularly at early disease stages and in oligosymptomatic cases [4,5].

In an effort to support a timely and accurate diagnosis, multiple imaging markers have been implemented. These morphometric MRI markers focus on the relatively selective midbrain and superior cerebellar peduncle (SCP) atrophy in RS, as evidenced by neuropathological studies [6]. Likewise, MSA (predominantly MSA-C and to a lesser extent MSA-P) is characterized by relatively selective pontine and middle cerebellar peduncle (MCP) atrophy [7]. Multiple studies have examined the diagnostic accuracy of diverse morphometric brainstem MRI markers, including linear distances, surfaces and volumes of the midbrain, the pons, SCPs, and MCPs [8–11]. Additionally, composite MRI markers, such as the magnetic resonance Parkinsonism indices 1 and 2 (MRPI 1 and 2)

have been introduced in an effort to increase discriminative power by combining multiple morphometrical measurements [12,13].

Despite the abundance of relevant studies, significant differences between studies in study designs, cohort characteristics, and imaging markers implemented have resulted in discrepant results regarding the diagnostic accuracy of these MRI markers in different APDs.

The present systematic review and meta-analysis aims to present data regarding MRI brainstem imaging markers in RS, MSA-P, and CBS in a systematic and comprehensive manner. The primary aim of this study was to compare the diagnostic accuracy of the most commonly applied MRI markers in cohorts of RS, MSA-P, and CBS, with a particular focus on planimetric and volumetric markers of midbrain and pons as well as composite MRI markers (MRPI 1 and 2).

## 2. Materials and Methods

The present study was performed according to the Preferred Reporting Items for Systematic Reviews and Meta-Analyses (PRISMA) statement. The study protocol was registered in the International Prospective Register for systematic reviews (PROSPERO; ID: CRD42023475739) [14]. No institutional board review approval was obtained since only previously published data were utilized.

### 2.1. Literature Search Strategy

PubMed and Scopus were searched from database inception to 15 October 2023 by three researchers independently (M.-E.B., I.K., and N.G.). In cases of disagreement regarding the eligibility of a study, these issues were discussed by all researchers and were included only after a consensus was reached. An additional manual search was performed on all included studies regarding: (a) all references of included studies; (b) all citations of these studies; (c) relevant studies (from PubMed). In cases of full text unavailability, the corresponding authors of papers were contacted in an effort to retrieve full texts.

The search strategy applied was: (MRI OR magnetic resonance OR brainstem OR midbrain OR pons OR cerebellar peduncle OR volume OR volumetry OR surface OR area OR planimetry OR distance OR diameter OR width) AND (CBD OR CBS OR corticobasal OR extrapyramidal OR gait apraxia OR MSA or multiple system atrophy OR parkinsonian OR parkinsonism OR PGAF or PSP or Richardson or supranuclear).

### 2.2. Eligibility Criteria and Study Selection

Inclusion criteria were:

(a)   Publication in English language;
(b)   Peer-reviewed, original research papers;
(c)   Studies with $\geq$ten patients in at least one patient group (RS, MSA-P or CBS) and $\geq$ten control subjects;
(d)   Studies including data on at least one of the following MRI markers: midbrain area ($M_{area}$), pons area ($P_{area}$), midbrain-to-pons-area ratio ($M/P_{area}$), midbrain volume ($M_{vol}$), pons volume ($P_{vol}$), midbrain-to-pons-volume ratio ($M/P_{vol}$), magnetic resonance Parkinsonism index 1 or 2 (MRPI 1 or MRPI 2, respectively);
(e)   Studies with available, extractable, or retrievable mean values and standard deviations (mean, SD) of MRI markers.

Exclusion criteria were:

(a)   Non-original studies (reviews, meta-analyses, case reports);
(b)   Studies with identical samples in different publications;
(c)   Studies including PSP cohorts of diverse phenotypes (e.g., RS, PSP-parkinsonism, primary gait apraxia with freezing) with unavailable or non-extractable data for the RS syndrome;
(d)   Studies including MSA cohorts of diverse phenotypes (i.e., MSA-cerebellar and MSA-parkinsonian) with unavailable or non-extractable data for MSA-P.

In cases of publications with possible partial overlap of cohorts, an algorithm including evaluation of authorship, study characteristics, sample characteristics, constructs' and measures' definitions, and study effects was applied to reach a decision regarding the eligibility of studies [15].

### 2.3. Data Extraction

Data extraction was performed by two authors independently (V.C.C. and N.G.). In cases of disagreement, a consensus was reached after joint assessment of the data from the original study.

Information extracted from studies included the following: first author; year of publication; study title; study design (i.e., retrospective, prospective, cross-sectional, unspecified); period of recruitment; study center.

Additionally, for each of the four groups included in this meta-analysis (RS, MSA-P, CBS, control group), the following information was extracted where available: male/female ratio; mean age; mean disease duration (applicable only in the patient groups). Also extracted were the subject count ($n$), mean value, and standard deviation (SD) of: (a) $M_{area}$; (b) $P_{area}$; (c) $M/P_{area}$; (d) $M_{vol}$; (e) $P_{vol}$; (f) $M/P_{vol}$: (g) MRPI 1; and (h) MRPI 2 per study group.

In cases of missing data on the $n$, mean, or SD of MRI markers, supplementary files of relevant papers were reviewed. Additionally, data were extracted from scatterplots, boxplot, or error bar plots where applicable through the use of WebPlotDigitizer version 4.6 (https://apps.automeris.io/wpd; access date: 15 November 2023).

### 2.4. Summary Measures

Standardized mean difference (SMD), as expressed by Cohen's $d$, was calculated to measure the effect size on the distinction between planimetric, volumetric, and composite MRI markers in APD patients and control subjects. Effect size based on Cohen's $d$ was interpreted as very small ($d \approx 0.01$), small ($d \approx 0.2$), medium ($d \approx 0.5$), large ($d \approx 0.8$), very large ($d \approx 1.2$), or huge ($d \approx 2.0$), based on recommendations [16].

### 2.5. Quality Evaluation

Quality evaluation was performed by three authors independently (M.-E.B., I.K., N.G.) through use of the QUADAS-2 tool [17]. It consists of four key domains—patient selection, index test, reference standard, and flow/timing—which are assessed in terms of bias and concerns regarding applicability. For the present meta-analysis, the signaling questions regarding the presence or absence of "pre-specified cut-offs" from the "reference standard" domain and the "appropriate interval between index test and reference standard" from the "flow and timing" domain were not implemented due to non-applicability. Additionally, the signaling question "Was a case–control design avoided?" from the "patient selection" domain was omitted since the primary aim of this meta-analysis was the comparison of MRI markers between APD patients and control subjects. Thus, per definition, all included studies had a case–control design. In cases of disagreement, a consensus was reached after discussions between the authors.

### 2.6. Statistical Analysis

The $Q$ statistic was used to assess the presence or absence of heterogeneity and the $I^2$ statistic was applied to quantify between-study heterogeneity. Heterogeneity was classified as low, moderate, or high with $I^2$ values of <25%, 25–50%, and >50%, respectively.

To control for between-study heterogeneity, a random effects model was applied for meta-analysis. Cohen's $d$ was calculated as a measure of the effect size of distinction between MRI markers in APD patients and control groups. Analyses were performed for each patient group (RA, MSA-P, and CBS) against the control group. No direct comparison between patient groups was performed. Forest plots were produced and displayed effect sizes, standard errors, confidence interval limits, $p$-values, and weights.

To test for publication bias, funnel plots were constructed with Cohen's d on the *x*-axis and standard error on the *y*-axis in order to visualize any outlying studies. Additionally, the Egger linear regression test was performed in order to quantify bias.

SPSS vs. 28 (IBM Corp. Released 2021; IBM SPSS Statistics for Windows, Version 28.0. Armonk, NY, USA: IBM Corp) was used by one author (V.C.C.) for all statistical analyses. A two-tailed *p* value < 0.05 was considered statistically significant.

## 3. Results

### 3.1. Literature Search and Screening Results

In total, 2076 were identified from PubMed and Scopus databases. After duplicate record elimination, 1096 records were screened by reviewing the title and abstract, eliminating 702 further studies. For the 337 remaining records, full texts were reviewed, eliminating 315 records (unrelated content: *n* = 280; patient or control group consisting of <10 subjects: *n* = 22 [18–37]; full text unavailable: *n* = 1 [38]; non-extractable data: *n* = 1 [39]; data in median/quartiles: *n* = 4 [40–43]; identical study cohorts or study cohorts with significant overlap: *n* = 2 [8,44]; data on PSP/MSA cohorts without distinction of RS/MSA-P phenotypes: *n* = 5 [45–49]). Five studies were identified via manual search of related papers, references, and citing papers [13,50–53]. A total of 27 studies were included in the systematic review and meta-analysis [10–13,50–71] (Figure 1).

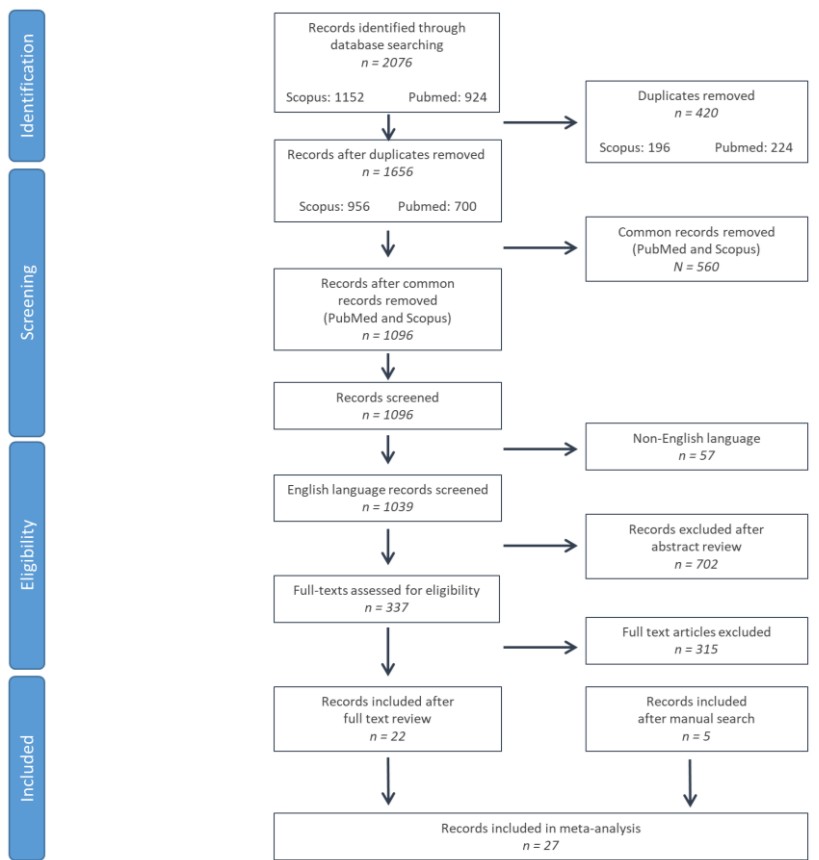

**Figure 1.** Flow chart of study selection according to preferred reporting items for systematic reviews and meta-analyses (PRISMA) criteria.

### 3.2. Basic Features Included in the Study

The basic characteristics of included studies are summarized in Table 1. Twenty-five studies included data on RS patients, five studies on MSA-P, and four studies on CBS. Eleven studies were retrospective, two studies were prospective, and the remaining had undefined study designs.

**Table 1.** Data regarding study design (Pr: prospective; Ret: retrospective; NC: not clarified), period of recruitment, male to female ratio, mean age (years), mean disease duration (months), and main findings of all studies included in the meta-analysis. NA: not available; dur: duration; ctrls: control subjects.

| | Study ID | Study Design | Period of Recruitment | PSP | | | | MSA | | | | CBS | | | | Main Findings |
|---|---|---|---|---|---|---|---|---|---|---|---|---|---|---|---|---|
| | | | | m/f | age | dur | data | m/f | age | dur | data | m/f | age | dur | data | |
| 1 | Oba, 2005 [10] | Pr | 1999–2003 | 1.6 | 71 | 33.6 | $M_{area}$; $P_{area}$; $M/P_{area}$ | 0.3 | 67 | 93.8 | $M_{area}$; $P_{area}$; $M/P_{area}$ | | | | (-) | $M/P_{area}$ differentiated RS from MSA-P |
| 2 | Groschel, 2006 [54] | Pr | 1997–1999 | 1.5 | 65.7 | 43.2 | $M_{area}$ | | | | (-) | 0.8 | 66.4 | 55.2 | $M_{area}$ | Decreased $M_{area}$ and $M/P_{area}$ values in RS vs. ctrls |
| 3 | Paviour, 2006 [55] | NC | NA | | 65.1 | 55.2 | $M_{vol}$; $P_{vol}$ | | 62.4 | 64.8 | (-) | | | | (-) | $M_{vol}$ 30% reduced in RS vs. ctrls |
| 4 | Cosottini, 2007 [11] | NC | NA | 1.1 | 72 | | $M_{area}$; $P_{area}$; $M/P_{area}$; $M_{vol}$ | 2.5 | 70 | | (-) | | | | (-) | Reduced $M_{area}$, $M/P_{area}$, $M_{vol}$ in RS vs. ctrls |
| 5 | Borroni, 2010 [56] | NC | 2001–2007 | 1.2 | 71.9 | 37.2 | $M/P_{area}$ | | | | (-) | 1.7 | 61.3 | 20.8 | $M/P_{area}$ | $M/P_{area}$ decreased in RS vs. CBS, ctrls |
| 6 | Longoni, 2011 [57] | Ret | 1998–2008 | 2.3 | 62.5 | 45.6 | $M_{area}$; $P_{area}$; $M/P_{area}$; MRPI 1 | | | | (-) | | | | (-) | MRPI 94% accuracy for RS vs. ctrls |
| 7 | Looi, 2011 [50] | NC | NA | 1.1 | 67.8 | | $M_{area}$; $P_{area}$; $M/P_{area}$ | | | | (-) | | | | (-) | $M_{area}$ differentiates RS vs. ctrls |
| 8 | Morelli, 2011 [12] | NC | NA | 2.8 | 70.3 | 42.8 | $M_{area}$; $P_{area}$; $M/P_{area}$; MRPI 1 | | | | (-) | | | | (-) | MRPI superior to $M.P_{area}$ in RS vs.ctrls |
| 9 | Morelli, 2014 [58] | NC | NA | 2.1 | 69 | 50.2 | $M_{area}$; $P_{area}$; $M/P_{area}$; MRPI 1 | | | | (-) | | | | (-) | MRPI superior to $M.P_{area}$ in RS vs.ctrls |
| 10 | Huppertz, 2016 [59] | NC | 2009–2013 | 1.3 | 69 | 38.4 | $M_{area}$; $P_{area}$; $M_{vol}$; $P_{vol}$ | 1.7 | 63.3 | 43.2 | $M_{area}$; $P_{area}$; $M_{vol}$; $P_{vol}$ | | | | (-) | $M_{vol}$ and $M_{area}$ decreased in RS |
| 11 | Mangesius, 2016 [60] | NC | NA | | | | (-) | 1.2 | 63.2 | 24.7 | $M_{area}$; $P_{area}$; $M/P_{area}$; MRPI 1 | | | | (-) | $M/P_{area}$ and MRPI comparable accuracy for RS |
| 12 | Pasha, 2016 [51] | NC | NA | 2.4 | 62.5 | 54 | $M_{area}$; $P_{area}$ | | | | (-) | | | | (-) | $M_{area}$ and $M/P_{area}$ differentiate RS vs. ctrls |
| 13 | Sankhla, 2016 [61] | NC | 2012–2014 | 2.2 | 66.1 | 30.7 | $M_{area}$; $P_{area}$; $M/P_{area}$; MRPI 1 | | | | (-) | | | | (-) | $M_{area}$, $M/P_{area}$, MRPI differentiate RS vs. ctrls |
| 14 | Nigro, 2017a [62] | NC | NA | 0.7 | 68.8 | 48.6 | $M_{area}$; $P_{area}$; MRPI 1 | | | | (-) | | | | (-) | MRPI >90% sens/ spec for RS vs. ctrls |

**Table 1.** *Cont.*

| | Study ID | Study Design | Period of Recruitment | PSP | | | | MSA | | | | CBS | | | | Main Findings |
|---|---|---|---|---|---|---|---|---|---|---|---|---|---|---|---|---|
| | | | | m/f | age | dur | data | m/f | age | dur | data | m/f | age | dur | data | |
| 15 | Nigro, 2017b [63] | NC | 2010–2016 | 2.0 | 71.0 | 45.6 | $M_{area}$; $P_{area}$; M/$P_{area}$; MRPI 1 | | | | (-) | | | | (-) | M/$P_{area}$, MRPI 100% sens/spec for RS vs. ctrls |
| 16 | Nizamani, 2017 [64] | Ret | 2006-2015 | 1.3 | 66.7 | 79.2 | $M_{area}$; $P_{area}$ | | | | (-) | | | | (-) | MRPI 100% sens/spec for RS vs. ctrls |
| 17 | Silsby, 2017 [65] | NC | NA | 0.6 | 71.1 | | $M_{area}$; $P_{area}$; M/$P_{area}$; MRPI 1 | | | | (-) | | | | (-) | M/$P_{area}$ and MRPI comparable AUCs in RS vs. ctrls |
| 18 | Quattrone, 2018 [13] | Ret | 2009–2017 | 1.2 | 70.4 | 46.8 | MRPI 1,2 | | | | (-) | | | | (-) | MRPI 1 and 2 identical AUCs in RS vs. ctrls |
| 19 | Ahn, 2019 [66] | Ret | 2010–2017 | 4.4 | 73 | 12.8 | $M_{area}$; $P_{area}$; M/$P_{area}$ | | | | (-) | | | | (-) | M/P ratio higher AUC compared to $M_{area}$ in RS vs. vtrls |
| 20 | Krismer, 2019 [67] | Ret | NA | | | | (-) | 1 | 59.7 | 32.4 | $M_{vol}$; $P_{vol}$ | | | | (-) | $P_{vol}$ did not differentiate MSA-P from ctrls |
| 21 | Quattrone, 2019 [68] | Ret | 2009–2017 | 1.2 | 70.9 | 48 | $M_{area}$ | | | | (-) | | | | (-) | $M_{area}$ decreased in RS vs. ctrls |
| 22 | Constantinides, 2019 [9] | Ret | 2012–2019 | 1.3 | 65.6 | 32.4 | $M_{area}$; M/$P_{area}$; MRPI 1 | | | | (-) | | | | (-) | $M_{area}$, M/$P_{area}$, MRPI decreased in RS vs. ctrls |
| 23 | Jabbari, 2020 [52] | Ret | 2015–2018 | | | | $M_{vol}$ | | | | (-) | | | | $M_{vol}$; $P_{vol}$ | $M_{vol}$ decreased in RS vs. ctrls |
| 24 | Nigro, 2020 [69] | Ret | 2010–2017 | 1.0 | 70.2 | 44.4 | $M_{area}$; $P_{area}$; MRPI 1 | | | | (-) | | | | (-) | Automated MRPI produces high diagnostic accuracy for RS |
| 25 | Madetko, 2022 [53] | NC | 2017–2019 | 1.7 | 74 | | $M_{area}$; $P_{area}$; M/$P_{area}$; MRPI 1,2 | 0.6 | 62.6 | | $M_{area}$; $P_{area}$; M/$P_{area}$; MRPI 1,2 | 0.1 | 72.8 | | $M_{area}$; $P_{area}$; M/$P_{area}$; MRPI 1,2 | $M_{area}$, M/$P_{area}$, MRPI 1,2 were decreased in RS vs. ctrls and to a lesser extent in CBS vs. ctrls |
| 26 | Virhammar, 2021 [70] | Ret | NA | 0.4 | | | $M_{area}$ | | | | (-) | | | | (-) | $M_{area}$ decreased in RS vs. ctrls |
| 27 | Quattrone, 2023 [71] | Ret | 2012–2020 | 1.0 | 70.7 | 46.8 | MRPI 1,2 | | | | (-) | | | | (-) | MRPI 1,2 produce AUC>0.975 in RS vs. ctrls |

Mean age in RS cohorts varied from 62.5–74 years, mean disease duration from 12.8–79.2 months, and male-to-female ratio from 0.4–4.4. In total, 21 studies included data on midbrain area, 17 on pons area, 11 on M/P$_{area}$, 13 on MRPI 1, 3 on MRPI 2, and 4 on midbrain volume (Table 1, Supplementary Table S1).

Mean age in MSA-P cohorts varied from 59.7–70 years, mean disease duration from 24.7–93.8 months, and male-to-female ratio from 0.3–2.5. Four studies included data on midbrain and pons area, and three studies included data on M/P$_{area}$ (Table 1, Supplementary Table S2).

None of the MRI markers had data on >2 studies for CBS cohorts. Mean age in CBS cohorts varied from 61.3–72.8 years, mean disease duration from 20.8–55.2 months, and male-to-female ratio from 0.1–1.7 (Table 1, Supplementary Table S3).

### 3.3. Quality Evaluation of Included Studies

Based on QUADAS-2 tool, the risk of bias regarding patient selection was unclear for 19 of the 27 studies included due to unspecified consecutive/random selection of patients in these studies. The risk of bias for the index test was unclear for 8 of the 27 studies and low for the remainder. The risk of bias for reference standard and flow/timing as well as concerns regarding applicability, was low for all included studies (Figure 2, Supplementary Table S1).

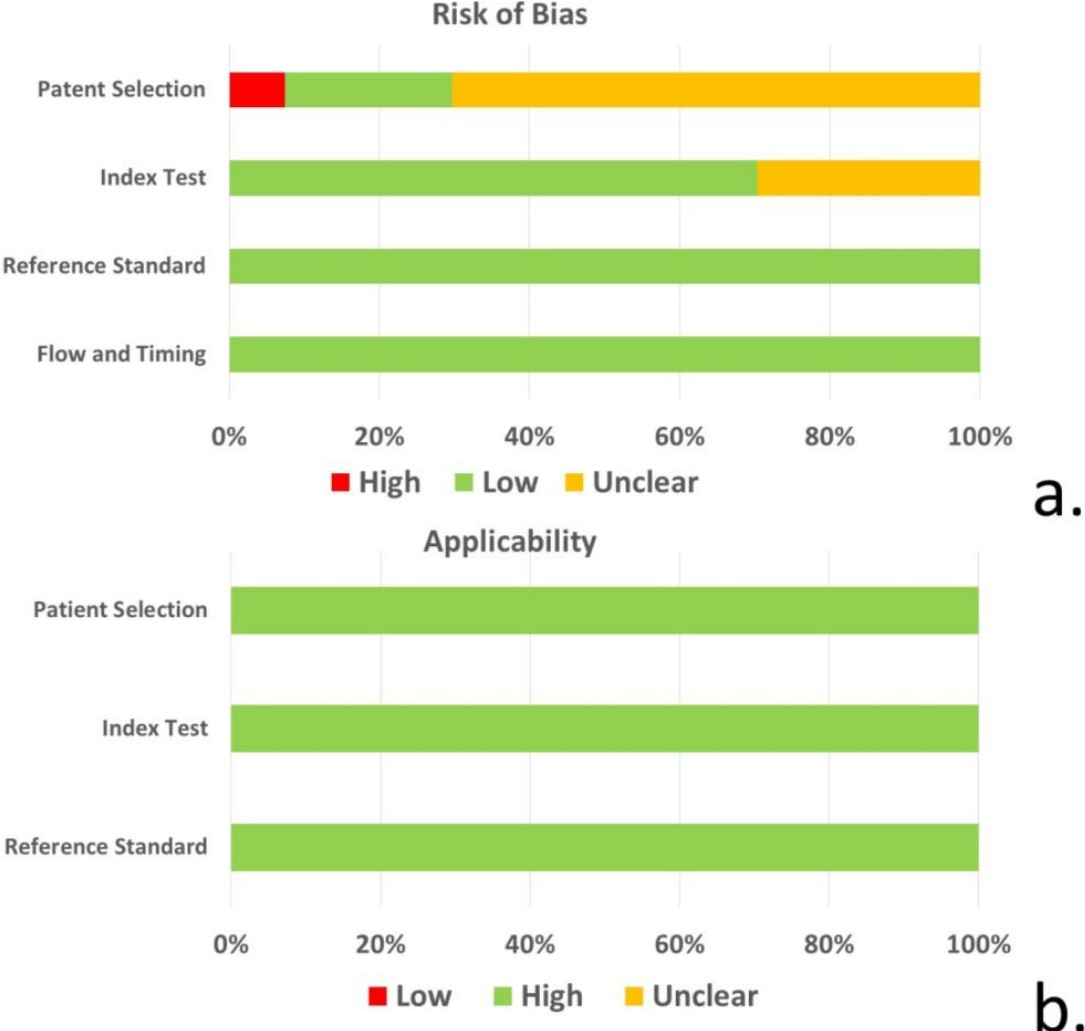

**Figure 2.** (**a**) Risk of bias and (**b**) concerns regarding the applicability of studies for patient selection, index test, reference standard, and flow/timing based on the QUADAS-2 tool.

### 3.4. Results of Meta-Analysis

3.4.1. Richardson's Syndrome

Twenty-one studies included data on midbrain area ($M_{area}$) in RS cohorts. A total of 1590 subjects (730 RS patients and 860 control subjects) were included in these studies. Mean midbrain area ranged from 59 mm$^2$ to 136.1 mm$^2$. M/F ratio ranged from 0.4 to 4.4. Mean age ranged from 62.5 to 74 years, and mean disease duration ranged from 12.8 to 79.2 months. All included studies reported significantly reduced midbrain surfaces, with Cohen's *d* ranging from −1.59 to −4.99. Overall Cohen's *d* for $M_{area}$ was −3.10 (−3.49 to −2.72; *p* < 0.001) (Table 1, Figure 3, Supplementary Table S2).

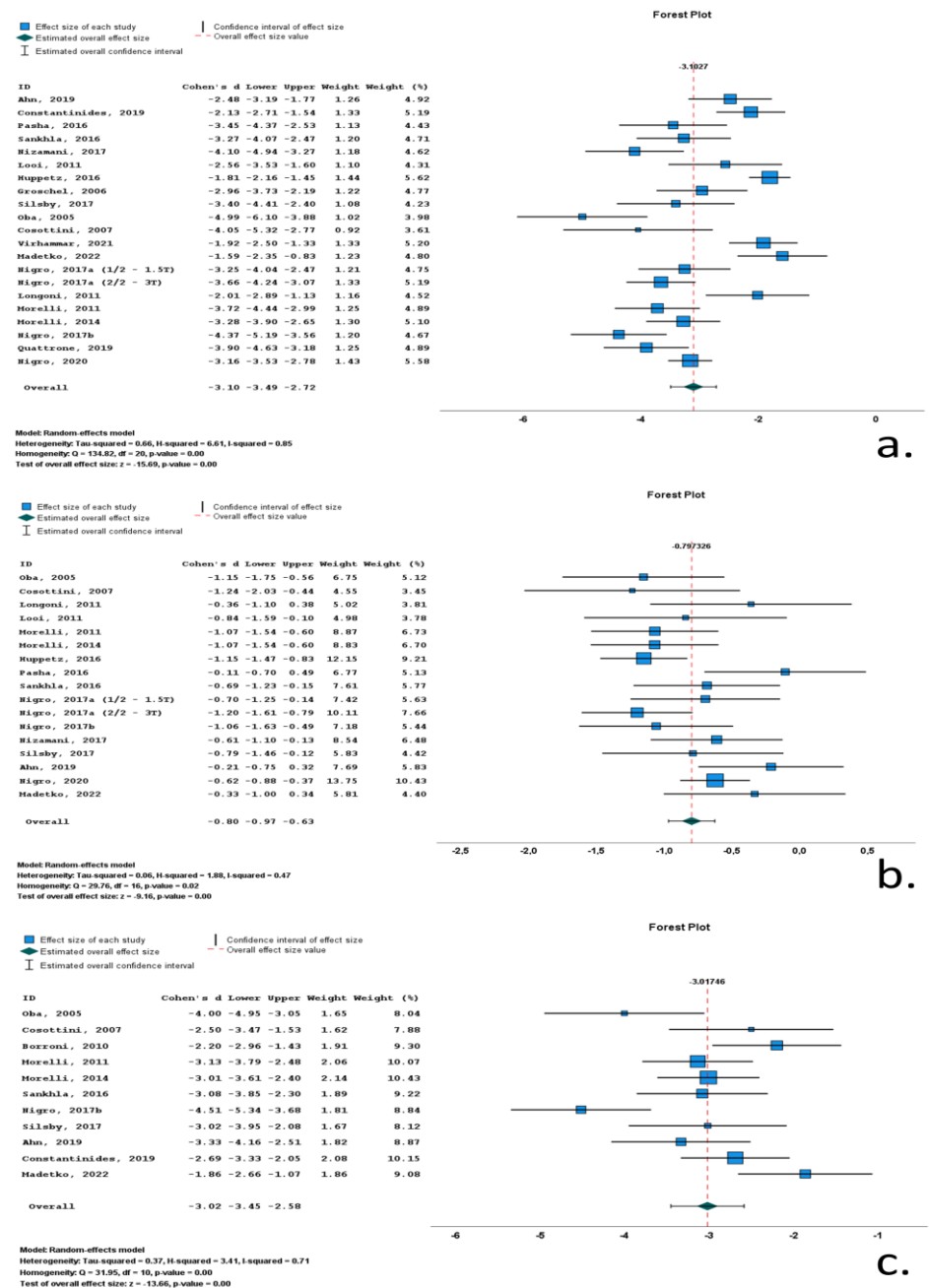

**Figure 3.** Forrest plots of effect size (as measured according to standardized mean difference-Cohen's *d*) of studies on Richardson's syndrome (RS) patients compared to control subjects for (**a**) midbrain area; (**b**) pons area; (**c**) midbrain-to-pons-area ratio. [9–12,50,51,53–59,61–70].

Seventeen studies included data on pons area ($P_{area}$) in RS cohorts. A total of 1348 subjects (577 RS patients and 771 control subjects) were included in these studies. Mean pons area ranged from 417 mm$^2$ to 526 mm$^2$. M/F ratio ranged from 0.6 to 4.4. Mean age ranged from 62.5 to 74 years, and mean disease duration ranged from 12.8 to 79.2 months. All included studies reported reduced pons surfaces, with Cohen's *d* ranging from −0.11 to −1.24. Overall Cohen's *d* for $P_{area}$ was −0.80 (−0.97 to −0.63; $p < 0.001$) (Table 1, Figure 3, Supplementary Table S2).

Eleven studies included data on midbrain-to-pons-area ratio (M/$P_{area}$) in RS cohorts. A total of 542 subjects (213 RS patients and 329 control subjects) were included in these studies. Mean M/$P_{area}$ ranged from 0.12 to 0.19. M/F ratio ranged from 0.6 to 4.4. Mean age ranged from 62.5 to 74 years, and mean disease duration ranged from 12.8 to 50.2 months. All included studies reported significantly reduced M/$P_{area}$, with Cohen's *d* ranging from −1.86 to −4.51. Overall Cohen's *d* for M/$P_{area}$ was −3.02 (−3.45 to −2.58; $p < 0.001$) (Table 1, Figure 3, Supplementary Table S2).

Thirteen studies included data on MRPI 1 in RS cohorts. A total of 1154 subjects (493 RS patients and 662 control subjects) were included in these studies. Mean MRPI 1 ranged from 17.6 to 27. M/F ratio ranged from 0.6 to 2.8. Mean age ranged from 62.5 to 74 years, and mean disease duration ranged from 30.7 to 50.2 months. All included studies reported significantly increased MRPI 1, with Cohen's *d* ranging from 1.35 to 6.94. Overall Cohen's *d* for MRPI 1 was 2.78 (2.05 to 3.52; $p < 0.001$) (Table 1, Figure 4, Supplementary Table S2).

Three studies included data on MRPI 2 in RS cohorts. A total of 229 subjects (127 RS patients and 102 control subjects) were included in these studies. Mean MRPI 2 ranged from 17.6 to 27. M/F ratio ranged from 1.0 to 1.7. Mean age ranged from 70.4 to 74 years, and mean disease was 46.8 months. All included studies reported significantly increased MRPI 2, with Cohen's *d* ranging from 1.87 to 3.11. Overall Cohen's *d* for MRPI 2 was 2.48 (1.80 to 3.172; $p < 0.001$) (Table 1, Figure 4, Supplementary Table S2).

Four studies included data on midbrain volume ($M_{vol}$) in RS cohorts. A total of 304 subjects (164 RS patients and 140 control subjects) were included in these studies. Mean $M_{vol}$ ranged from 17.6 to 27. M/F ratio ranged from 1.1 to 1.3. Mean age ranged from 65.1 to 72 years, and mean disease ranged from 38.4 to 55.2 months. All included studies reported significantly decreased $M_{vol}$, with Cohen's *d* ranging from −1.51 to −2.74. Overall Cohen's *d* for $M_{vol}$ was −1.99 (−2.27 to −1.71; $p < 0.001$) (Table 1, Figure 4, Supplementary Table S2).

### 3.4.2. MSA-P

Four studies included data on midbrain area ($M_{area}$) in MSA-P cohorts. A total of 275 subjects (126 MSA-P patients and 149 control subjects) were included in these studies. Mean $M_{area}$ ranged from 97.2 mm$^2$ to 153.8 mm$^2$. M/F ratio ranged from 0.3 to 1.7. Mean age ranged from 62.6 to 67 years and mean disease ranged from 24.7 to 93.8 months. All included studies reported significantly decreased $M_{area}$, with Cohen's *d* ranging from −0.55 to −1.30. Overall Cohen's *d* for $M_{area}$ was −0.97 (−1.34 to −0.59; $p < 0.001$) (Table 1, Figure 5, Supplementary Table S3).

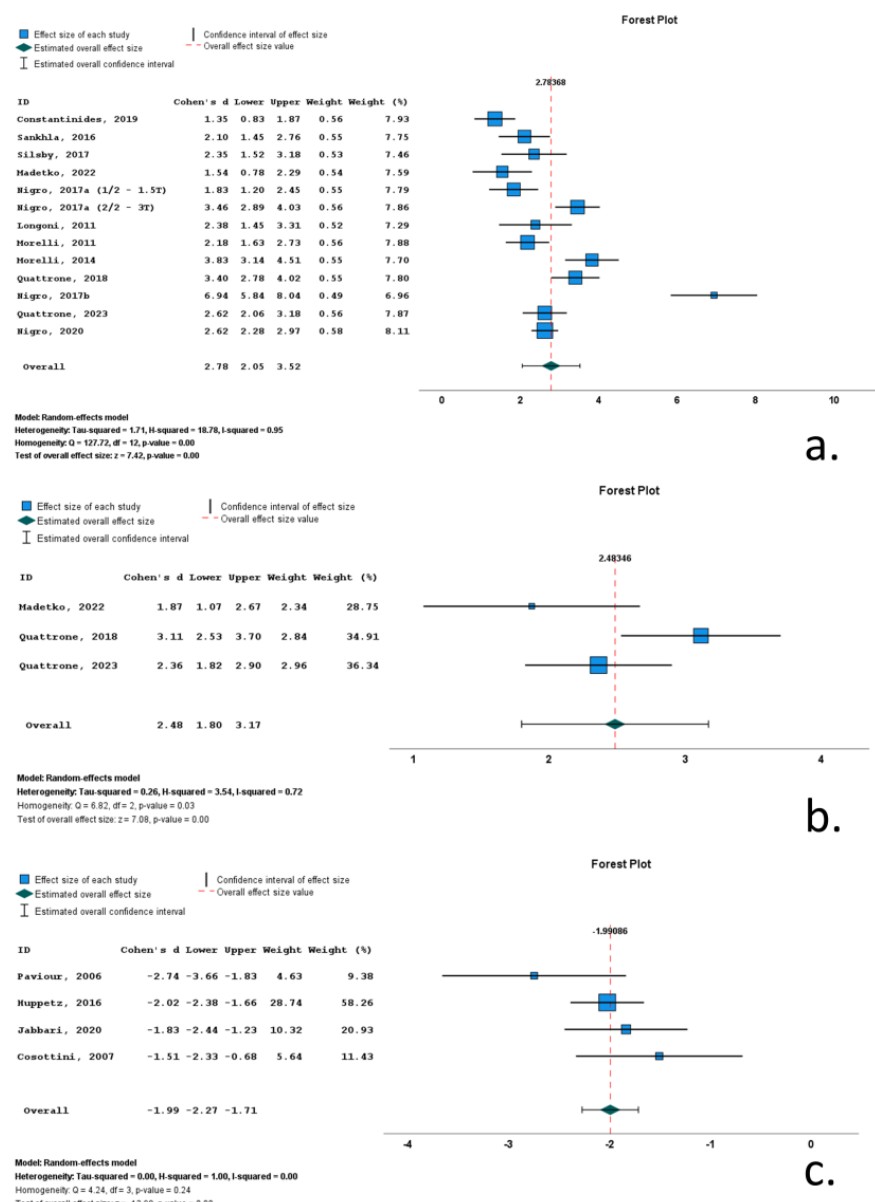

**Figure 4.** Forrest plots of effect size (as measured by standardized mean difference—Cohen's *d*) of studies on Richardson's syndrome patients compared to control subjects for: (**a**) MRPI 1; (**b**) MRPI 2; (**c**) midbrain volume. [9,11–13,52,53,55,57–59,61–63,65,68,69,71].

Four studies included data on pons volume (P$_{area}$) in MSA-P cohorts. A total of 275 subjects (126 MSA-P patients and 149 control subjects) were included in these studies. Mean P$_{area}$ ranged from 381.6 mm$^2$ to 459 mm$^2$. M/F ratio ranged from 0.3 to 1.7. Mean age ranged from 62.6 to 67 years, and mean disease ranged from 24.7 to 93.8 months. All included studies reported significantly decreased P$_{area}$, with Cohen's *d* ranging from −1.68 to −0.54. Overall Cohen's *d* for P$_{area}$ was −1.15 (−1.57 to −0.72; *p* < 0.001) (Table 1, Figure 5, Supplementary Table S3).

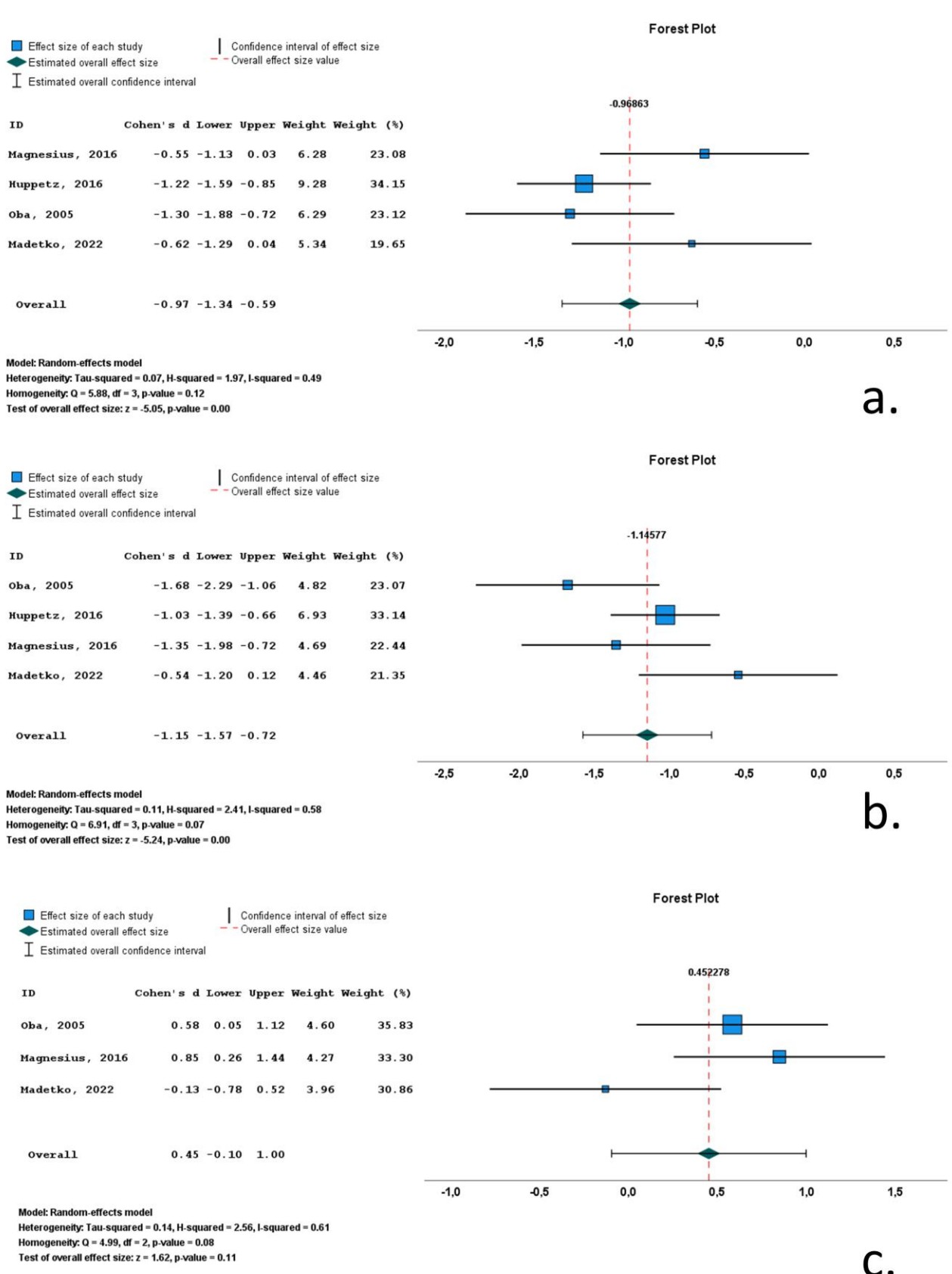

**Figure 5.** Forrest plots of effect size (as measured by standardized mean difference—Cohen's *d*) of studies on MSA-P patients compared to control subjects for: (**a**) midbrain area; (**b**) pons area; (**c**) midbrain-to-pons-area ratio. [10,53,59,60].

Three studies included data on midbrain-to-pons-area ratio (M/P$_{area}$) in MSA-P cohorts. A total of 142 subjects (66 MSA-P patients and 76 control subjects) were included in these studies. Mean M/P$_{area}$ ranged from 0.24 to 0.27. M/F ratio ranged from 0.1 to 1.2. Mean age ranged from 62.6 to 67 years, and mean disease ranged from 24.7 to 93.8 months. Two of the included studies reported significantly increased M/P$_{area}$, whereas a single study reported a decreased M/P$_{area}$, with Cohen's *d* ranging from −0.13 to 0.85. Overall Cohen's *d* for M/P$_{area}$ was 0.45 (−0.10 to 1.00; *p* = 0.11) (Table 1, Figure 5, Supplementary Table S3).

### 3.4.3. CBS

Meta-analysis could not be performed because none of the MRI markers had available data in >2 studies (Supplementary Table S4).

### *3.5. Heterogeneity*

The *Q* statistic was used to assess the presence or absence of heterogeneity qualitatively, and the $I^2$ statistic was applied to quantify between-study heterogeneity.

In the RS studies, midbrain area (*Q* statistic *p*-value < 0.001; $I^2$ = 0.85), midbrain-to-pons-area ratio (*Q* statistic *p*-value < 0.001; $I^2$ = 0.71), MRPI 1 (*Q* statistic *p*-value < 0.001; $I^2$ = 0.95), and MRPI 2 (*Q* statistic *p*-value < 0.001; $I^2$ = 0.72) exhibited high heterogeneity. Pons area exhibited moderate heterogeneity (*Q* statistic *p*-value = 0.02; $I^2$ = 0.47), and midbrain volume did not exhibit heterogeneity (*Q* statistic *p*-value = 0.24; $I^2$ = 0.00)

In the MSA-*p* studies, midbrain area had moderate heterogeneity (*Q* statistic *p*-value = 0.12; $I^2$ = 0.49), whereas pons area (*Q* statistic *p*-value = 0.07; $I^2$ = 0.58) and midbrain-to-pons-area ratio (*Q* statistic *p*-value = 0.08; $I^2$ = 0.61) had high heterogeneity.

### *3.6. Publication Bias*

Funnel plots indicated a degree of asymmetry in RS patients for midbrain area and MRPI 1, indicative of publication bias. The results of the Egger's-regression-based test did not support the presence of publication bias for MRPI 1 in RS but confirmed publication bias for midbrain area in RS ($\beta_0$ = −1.639; −3.034—0.244; *p*-value = 0.024) and pons area in RS ($\beta_0$ = −1.045; −1.735—0.355; *p*-value = 0.006) (Figure 6).

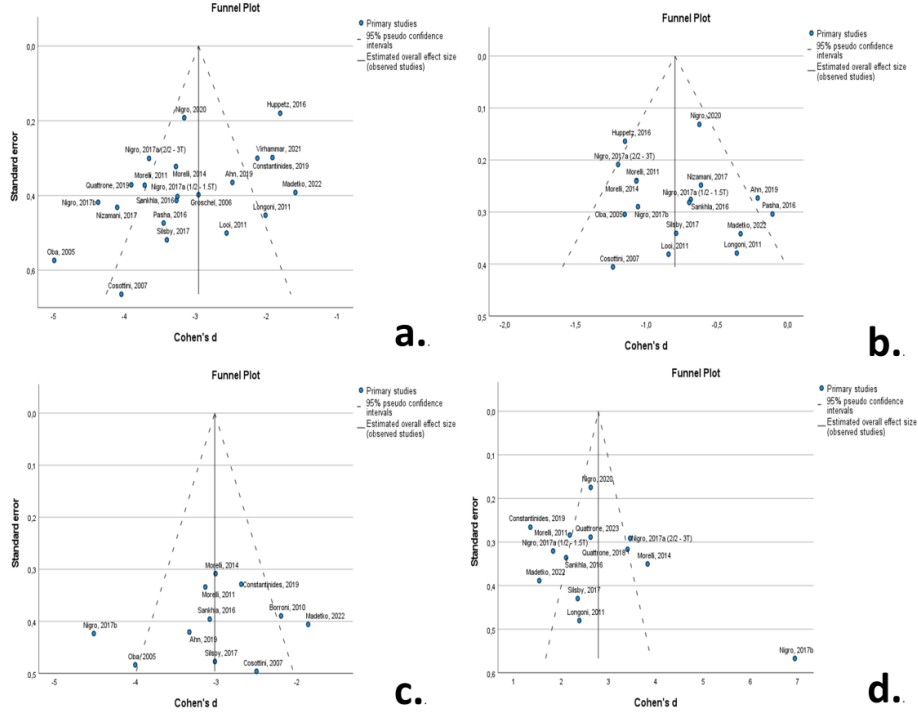

**Figure 6.** *Cont.*

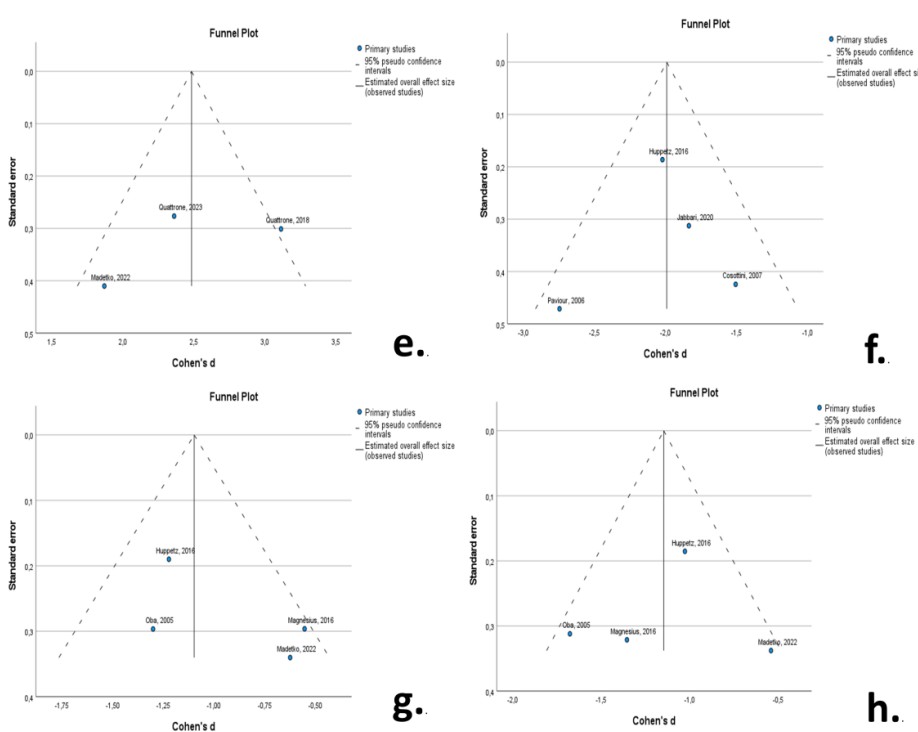

**Figure 6.** Funnel plots to visualize possible publication bias for: (**a**) midbrain area in RS; (**b**) pons area in RS; (**c**) M/P$_{area}$ in RS; (**d**) MRPI 1 in RS; (**e**) MRPI 2 in RS; (**f**) midbrain volume in RS; (**g**) midbrain area in MSA-P; (**h**) pons area in MSA-P. A degree of asymmetry is evident for midbrain area in RS (**a**) and MRPI in RS (**d**). [9–13,50–66,68–71].

## 4. Discussion

Progressive supranuclear palsy, multiple system atrophy, and corticobasal degeneration are rare neurodegenerative Parkinsonian disorders with characteristic neuropathological features which present with multiple diverse phenotypes. Richardson's syndrome (RS) is the prototypical manifestation of PSP and is characterized by early postural instability and supranuclear gaze palsy [1]. MSA-Parkinsonism (MSA-P) manifests as predominant Parkinsonism combined with dysautonomia, cerebellar, and pyramidal signs [2]. Corticobasal syndrome (CBS) manifests with symptoms and signs of cortical (apraxia, cortical sensory deficits, alien limb phenomena) and basal ganglionic dysfunction (parkinsonism, myoclonus, dystonia) [3]. Despite the presence of these unique clinical features, patients with APD are commonly misdiagnosed, particularly early in their disease course as well as in oligosymptomatic or atypical/mixed presentations.

In an effort to improve a timely and accurate diagnosis of APD, multiple imaging markers have been introduced. These markers range from morphometric MRI markers—which are clinically applicable—to more elaborate, research-oriented markers, including diffusion, resting-state MRI, SPECT, and PET markers [72–84].

Neuropathological studies have supported that PSP is characterized by preferential midbrain and SCP atrophy, whereas MSA (particularly MSA-C) is characterized by pontine and MCP atrophy. To this end, most morphometric MRI markers in APD have focused on midbrain, pons, as measured through midbrain and pons areas and volumes. Additionally, composite markers such as MRPI 1 and MRPI 2 have been introduced, which incorporate measurements of midbrain and pons surfaces as well as SCP and MCP widths.

Multiple MRI studies have focused on the planimetric and volumetric midbrain/pons characteristics of RS, MSA-P, and CBS. However, these studies exhibit differences in design, diagnostic criteria, patient characteristics/groups, and imaging markers applied. In order to systematically present data on these markers, we performed a systematic review of all studies on RS, MSA-P, and CBS which included at least one of the following imaging

markers: midbrain area and/or volume, pons area and/or volume, midbrain-to-pons-area and/or volume ratio, and MRPI 1 and 2.

An initial conclusion of the present systematic review is that few studies have applied these MRI markers in MSA-P ($n$ = 5) or CBS ($n$ = 4). Meta-analysis could not be performed for any of the MRI markers in CBS because none of these markers had available data in >2 studies. For MSA-p, only three MRI markers (midbrain area, pons area and M/P$_{area}$) had available data on >2 studies and were thus available for meta-analysis. Based on Cohen's $d$ as a measure of effect size as measured by pons area, MSA-p patients present with predominant pontine atrophy (Cohen's $d$ = −1.15; $p < 0.001$). However, these patients also exhibit comparable midbrain atrophy (Cohen's $d$ = −0.97; $p < 0.001$), thus rendering the M/P$_{area}$ an ineffective surrogate marker for MSA-P. Thus, pontine atrophy, as measured by pons surface in the midsagittal plane, is the most potent MRI marker for MSA-P.

Twenty-five studies included data on MRI markers in RS. Meta-analysis was performed for all MRI markers, except for pons volume and M/P$_{vol}$, due to lack of >2 studies with data. Midbrain area provided the greatest Cohen's $d$ value among MRI markers (Cohen's $d$ = −3.10; $p < 0.001$), followed by M/P$_{area}$ (Cohen's $d$ = −3.02; $p < 0.001$), MRPI 1 (Cohen's $d$ = 2.78; $p < 0.001$) and MRPI 2 (Cohen's $d$ = 2.48; $p < 0.001$). The greater effect size of midbrain area compared to M/P$_{area}$, MRPI 1 and MPRI 2 could be attributed to the concomitant pontine atrophy in RS (as evidenced by pons area Cohen's $d$ = 0.80; $p = 0.02$). These data indicate that despite the introduction of composite MRI markers such as the MRPI, measurement of the midbrain surface remains the most effective MRI marker for PSP.

The level of evidence, based on the number of subjects included per analysis, varied between RS studies, with midbrain area ($n$ = 1590), pons atrophy ($n$ = 1348), and MRPI 1 ($n$ = 1154) included in the largest samples. Meta-analysis for MSA-P studies included significantly smaller samples ($n$ = 275) for midbrain and pons areas. Publication bias was present for midbrain area studies in RS, and heterogeneity among studies was high for multiple MRI markers.

There are certain limitations to this systematic review and meta-analysis. Initially, most studies included had positive results. We did not perform a systematic search of the grey literature, to search for negative relevant unpublished studies. However, publication bias based on funnel plots and Egger's-regression-based test was minimal. Additionally, negative studies on pontine area in RS and on midbrain area, pons area, and M/P$_{area}$ were published and included. Lastly, the effect size for most MRI markers in RS in the included studies was consistently high, rendering the possibility of negative relevant studies unlikely. Another limitation of this study was the inclusion of all relevant MRI studies with planimetric/volumetric brainstem data irrespective of the methodology used (i.e., planimetry methodology based on Oba et al. vs. Cossotini et al. [10,11]; inclusion or exclusion of midbrain tectum [59]; manual vs. automatic measurements [63]; 1.5 T vs. 3 T MRI [60]). The variability in these factors may have contributed to the heterogeneity of studies. However, relevant studies have supported excellent agreement between automated and manual measurements and between MRI scanners of different field strengths. Only two of the included studies had a prospective design, with most studies being either retrospective or undefined. Thus, the temporal pattern of atrophy based on MRI markers of this meta-analysis cannot be deduced. The studies included in this meta-analysis applied different MRI acquisition protocols with varied TR, TE, FOV, and slice thickness values (Supplementary Table S5). Differences in resulting planimetric or volumetric measurements due to these MRI acquisition protocol differences could not be tested due to the great variability between studies. Rarely, midline structural lesions, such as vascular malformations, tumors or traumatic lesions may result in phenotypes mimicking atypical Parkinsonism. All studies included in this meta-analysis excluded patients with structural lesions. Application of the MRI markers discussed in this review implies the absence of such lesions. Lastly, all included studies rely on the classification of patients based on established clinical diagnostic criteria, in lack of neuropathological confirmation, rendering misdiagnosis of patients possible.

Prospective longitudinal studies of large cohorts of RS, MSA-P, and CBS patients, with neuropathological confirmation, would be pivotal in elucidating the temporal and spatial patterns of brainstem structure atrophy and the optimal surrogate MRI markers for these rare diseases.

**Supplementary Materials:** The following supporting information can be downloaded at: https://www.mdpi.com/article/10.3390/neurolint16010001/s1, Supplementary Table S1: "Analytical presentation of study characteristics regarding the signaling questions, risk of bias and concerns regarding applicability for patient selection, Index Test, Reference Standard and Flow/Timing, based on the QUADAS Tool"; Supplementary Table S2: "Analytical data regarding the number of patients per study group, mean values and standard deviations of all available MRI markers in studies of Richardson patients included in the meta-analysis"; Supplementary Table S3: "Analytical data regarding the number of patients per study group, mean values and standard deviations of all available MRI markers in studies of MSA-P patients included in the meta-analysis"; Supplementary Table S4: "Analytical data regarding the number of patients per study group, mean values and standard deviations of all available MRI markers in studies of corticobasal syndrome patients included in the meta-analysis"; Supplementary Table S5: "MRI acquisition protocol per study".

**Author Contributions:** Conceptualization: M.-E.B., I.K., N.G. and V.C.C.; methodology: V.C.C.; formal analysis: V.C.C.; investigation: M.-E.B., I.K., N.G. and V.C.C.; data curation: M.-E.B., I.K., N.G. and V.C.C.; writing—original draft preparation: V.C.C.; writing—review and editing: M.-E.B, I.K. and N.G.; supervision, V.C.C. All authors have read and agreed to the published version of the manuscript.

**Funding:** This research received no external funding.

**Institutional Review Board Statement:** The study design of this paper (systematic review and meta-analysis) does not require institutional review board approval.

**Informed Consent Statement:** This study did not include original data. The study design (systematic review and meta-analysis) does not require informed consent.

**Data Availability Statement:** Data is contained within the article or supplementary material.

**Conflicts of Interest:** The authors declare no conflict of interest.

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
