# Peer review of "Planimetric and Volumetric Brainstem MRI Markers in Progressive Supranuclear Palsy, Multiple System Atrophy, and Corticobasal Syndrome. A Systematic Review and Meta-Analysis"

_2035-8377, doi:10.3390/neurolint16010001_

Round 1

Reviewer 1 Report

Comments and Suggestions for Authors

The article by Brinia M.E. et al. titled as “Planimetric and volumetric brainstem MRI markers in progressive supranuclear palsy, multiple system atrophy and corticol basal syndrome. A systematic review and meta-analysis” highlight the recent development and the role of imaging techniques in the accurate diagnosis of Atypical parkinsonian disorders. The usefulness of MR planimetric quantitative biomarkers to differentiate neurological disorders has been discussed by the authors. This is an interesting manuscript, and the topic is highly relevant. However, there are several points need to be addressed to improve the quality of the manuscript.

-The article is based on MRI T1 images but the T1 protocol is not mentioned. What type of acquisition was used? Volumetric (3D), or T1 in 3 different planes. If a non-volumetric acquisition was used, what is the slice thickness, gap, etc. These parameters can influence the final measurements.

-All figures are not clear, please provide high resolutions images.

Author Response

Reviewer 1

Comment 1: “The article is based on MRI T1 images but the T1 protocol is not mentioned. What type of acquisition was used? Volumetric (3D), or T1 in 3 different planes. If a non-volumetric acquisition was used, what is the slice thickness, gap, etc. These parameters can influence the final measurements”.

Reply: Thank you for your insightful remark. Studies have supported excellent agreement between MRI scanners of different field strengths (i.e. 1.5T vs. 3T) and between automated and manual measurements. Regarding the T1 MRI sequence characteristics, these vary among studies, since every center used different MRI scanners with different T1 sequences. Thus a variety of slice thickness,FOV, TR and TE values were implemented. These data were added on Supplemetary Table 5:

3D

Field strength

Slice thickness

FOV

Imaging matrix

TR

TE

Oba, 2005

1.5T

3.0mm-4.0mm

24cm

256 x 192

500

9

Groschel, 2006

(+)

1.5T

2.0mm

256 x 192

24

5

Paviour, 2006

1.5T

1.5mm

24 x 18cm

256 x 256

13

5.4

Cosottini, 2007

(+)

1.5T

1.2mm

25cm

256 x 256

14.4

6.9

Borroni, 2010

1.5T

Longoni, 2011

(+)

1.5T

1.0mm

23.6 x 23.6 cm

256 x 224 x208

2000

4.72

Looi, 2011

3.0T

1.0mm

24cm

240 x 240

8

4

Morelli, 2011

(+)

1.5T

0.6mm

256 x 256

15.2

6.8

Morelli, 2014

(+)

1.5T

0.6mm

256 x 256

15.2

6.8

Huppertz, 2016

(+)

1.5T / 3.0T

1.0mm

Mangesius, 2016

(+)

1.5T

1.2mm

22 x 16.5cm

256x204

1800

2.18

Pasha, 2016

3T

Sankhla, 2016

(+)

1.5T / 3.0T

Nigro, 2017a

(+)

1.5T / 3.0T

Nigro, 2017b

(+)

3.0T

1.0mm

9.2

3.7

Nizamani, 2017

(+)

1.5T / 3.0 T

0.6mm

256 x 256

15.2

6.8

Silsby, 2017

(+)

3.0T

1.0mm

256 x 256

5.8

2.6

Quattrone, 2018

(+)

3.0T

1.0mm

25.6cm

256 x 256

9.2

3.7

Ahn, 2019

Krismer, 2019

(+)

3.0T

1.2mm

22 x 16.5cm

256 x 204

1800

2.18

Quattrone, 2019

(+)

3.0T

1.0mm

25.6cm

256 x 256

9.2

3.7

Constantinides, 2020

(+)

1.5T / 3.0 T

0.6mm – 2.0mm

24-25cm

1960 - 9092

1.8-4.6

Jabbari, 2020

(+)

3.0T

1.1mm

2.93

2

Nigro, 2020

(+)

1.5T / 3.0 T

1.0mm – 1.2mm

6.7 – 1800

2.19 – 4.04

Madetko, 2022

3.0T

Virhammar, 2022

(+)

3.0T

1mm

Quattrone, 2023

(+)

3.0T

1mm

9.2

3.7

Supplementary Table 5. MRI acquisition protocol per study

The following sentences were added in the Discussion section:

“The studies included in this meta-analysis applied different MRI acquisition protocols, with varied TR, TE, FOV and slice thickness values (Supplementary Table 5). Differences in resulting planimetric or volumetric measurements due to these MRI acquisition protocol differences could not be tested, due to the great variability among studies”.

Comment 2: “All figures are not clear, please provide high resolutions images”.

Reply: Thank you for your comment. High resolution images were provided.

Reviewer 2 Report

Comments and Suggestions for Authors

Thank you for inviting me to review this manuscript.

In this systematic review/meta-analsis the authors aim to compare the diagnostic accuracy of various MRI markers, including midbrain and pons areas and volumes, ratios and composite markers (Magnetic Resonance Imaging Parkinsonism Indices; MRPI that have been proposed as imaging markers of Richardson’s syndrome (RS) and multiple system atrophy-parkinsonism (MSA-p).

I have enjoyed reading this well-written manuscript, the topic is of interest, the methods are well documented and are sounding to me, the work seems well performed and sufficiently detailed. I have only few concerns:

·      The discussion seems to be a repetition of the results, the exposition of a reflection on what the authors find is missing. Furthermore, there seems to be a lack of a take-home message that gives meaning to the work.

·      the resolution of the figures and tables is low and it is not possible to read them.

Comments on the Quality of English Language

Minor editing of English language required

Author Response

Reviewer 2

Comment 1: “The discussion seems to be a repetition of the results, the exposition of a reflection on what the authors find is missing. Furthermore, there seems to be a lack of a take-home message that gives meaning to the work”.

Reply: Thank you for this comment. We have added the following sentences in the Discussion section to provide take-home messages to the reader:

“Thus, pontine atrophy, as measured by pons surface in the midsagittal plane is the most potent MRI marker for MSA-P”.

AND

“These data indicate that despite the introduction of composite MRI markers such as the MRPI, measurement of the midbrain surface remains the most effective MRI marker for PSP”.

Comment 2: “The resolution of the figures and tables is low and it is not possible to read them”.

Reply: Thank you for your remark. High resolution images were added.

Reviewer 3 Report

Comments and Suggestions for Authors

Authors present a systematic review and meta-analysis on magnetic resonance imaging (MRI) markers in progressive supranuclear palsy, multiple system atrophy and corticobasal syndrome. Methods, results and conclusions appear convincing. The discussion section of the manuscript might benefit from some thoughts as to the detection, quantification and therapy of non-degenerative lesions in midline structures of the brain, e.g., cavernous malformations, tumours, and traumatic lesions.

Comments on the Quality of English Language

Minor spell check and minor changes in wording recommended.

Author Response

Reviewer 3

Comment 1: “The discussion section of the manuscript might benefit from some thoughts as to the detection, quantification and therapy of non-degenerative lesions in midline structures of the brain, e.g., cavernous malformations, tumours, and traumatic lesions”.

Reply: Thank you for your insightful remark. The main aim of this meta-analysis was to provide information on the literature regarding brainstem morphological changes based on MRI, of patients with atypical Parkinsonism. To this end, all included studies had excluded patients with structural lesions such as tumors, traumatic lesions or cavernous malformations, which may sometimes result in atypical parkinsonian syndromes.

We have added the following sentence in the Discussion section:

“Rarely, midline structural lesions, such as vascular malformations, tumors or traumatic lesions may result in phenotypes mimicking atypical Parkinsonism. All studies included in this meta-analysis excluded patients with structural lesions. Application of the MRI markers discussed in this review imply the absence of such lesions”.